# Cumulative Effective Dose from Medical Imaging in Inflammatory Bowel Disease

**DOI:** 10.3390/diagnostics11122387

**Published:** 2021-12-18

**Authors:** Agata Łukawska, Dominika Ślósarz, Aneta Zimoch, Karol Serafin, Elżbieta Poniewierka, Radosław Kempiński

**Affiliations:** 1Department of Gastroenterology and Hepatology, Wroclaw Medical University, 213 Borowska, 50–556 Wroclaw, Poland; dominika.slosarz@student.umw.edu.pl (D.Ś.); elzbieta.poniewierka@umw.edu.pl (E.P.); radoslaw.kempinski@umw.edu.pl (R.K.); 2Faculty of Medicine, Wroclaw Medical University, 50-367 Wroclaw, Poland; anecia1304@wp.pl (A.Z.); seraf.karol@gmail.com (K.S.)

**Keywords:** cumulative effective dose, imaging studies, inflammatory bowel disease

## Abstract

Inflammatory bowel diseases (IBD) are chronic and relapsing disorders usually requiring numerous medical imaging. IBD patients might be exposed to a large dose of radiation. As a cumulative effective dose (CED) ≥ 50 mSv is considered significant for stochastic risks of cancer, it is important to monitor the radiation exposure of IBD patients. In the present work, we aimed to quantify the mean CED in IBD patients and identify factors associated with exposure to high doses of diagnostic radiation. A retrospective chart view of patients with IBD hospitalized between 2015 and 2019 was performed. A total of 65 patients with Crohn’s disease (CD) and 98 patients with ulcerative colitis (UC) were selected. Of all imaging studies performed, 73% were with doses of ionizing radiation. Mean CED (SD) amounted to 19.20 (15.64) millisieverts (mSv) and 6.66 (12.39) mSv, respectively, in patients with CD and UC (*p* < 0.00001). Only 1.84% of the patients received CED ≥ 50 mSv. We identified three factors associated with CED in the IBD patients: number of surgical procedures, and number and length of hospitalization. CD patients with strictures or penetrating disease and UC patients with extensive colitis were more likely to receive higher radiation doses.

## 1. Introduction

Inflammatory bowel diseases (IBD) are chronic, progressive disorders that include Crohn’s disease (CD) and ulcerative colitis (UC). The first symptoms of the disease usually occur in the second to fourth decade of life. IBD is characterized by periods of remission and exacerbations. Patients with IBD require frequent diagnostic tests, which are performed in order to state the diagnosis, estimate the extent of the disease, determine the patient’s response to the treatment, monitor disease activity, and evaluate complications [1,2]. Endoscopic examination is a gold standard for the diagnosis and evaluation of disease activity but has limitations. Colonoscopy is an invasive procedure and requires difficult preparation [3]. The mucosa and the lumen of the gut can be easily described but the deeper layers of the wall cannot be evaluated.

Increasingly, imaging studies are performed. Magnetic resonance imaging (MRI), ultrasound (US), X-ray, and computed tomography (CT) are used to evaluate changes in the intestine and detect complications, such as abscesses, fistula, or bowel obstruction. When there are contraindications for the colonoscopy, or the examination is impossible because of the strictures of the intestine or adhesions in the abdominal cavity, imaging studies can be considered as well. With the exception of MRI and US, these studies expose patients to ionizing radiation. Multiple imaging tests are related to higher doses of ionizing radiation exposure [4].

Radiation exposure is expressed as an effective dose of radiation, which is measured in millisieverts (mSv). Value of dose depends on the type of radiation, body part’s sensitivity to radiation, age, gender, and the radiologist’s protocol [5]. Doses of radiation above 100 mSv are related to a well-documented increased cancer mortality risk. Some data indicate that a range of doses between 10 and 100 mSv may also increase cancer risk [6]. The threshold defined as a potentially harmful diagnostic radiation exposure is commonly defined as a cumulative effective dose (CED) ≥ 50 mSv [4]. IBD patients are at risk of developing colorectal and small intestinal cancers, especially in long lasting disease. Furthermore, immunosuppressive drugs increase the risk of melanoma, nonmelanoma skin cancer, and lymphoma among patients with IBD. Additionally, exposure to high doses of ionizing radiation increases the risk of developing cancer [2,7,8].

In the present work, we aimed to quantify the mean CED in patients with IBD to assess any harmful effects of radiation and identify factors associated with exposure to high doses of diagnostic radiation.

## 2. Materials and Methods

### 2.1. Study Design and Population

A total of 1791 consecutive hospital charts of patients with IBD hospitalized in the Gastroenterology and Hepatology Department of Wroclaw Medical University between January 2015 and December 2019 were meticulously searched. A total of 163 adults, who were admitted to the hospital at least three times, were included in the study, 65 patients with CD and 98 patients with UC. Two groups were statistically homogenous. The flowchart for patient inclusion is presented in Figure 1. UC patients were divided into subgroups due to the disease extension: E1—ulcerative proctitis, E2—left-sided colitis and E3—extensive colitis (changes proximal to splenic flexure). CD patients were divided due to the disease location: L1—ileal, L2—colonic, L3—ileocolonic and behavior of the disease: B1—non-stricturing/non-penetrating, B2—stricturing, B3—penetrating. We conducted an analysis of the onset and duration of the disease, underwent pharmacotherapy, and the hospitalization days spent in the Gastroenterology and Hepatology Department. Furthermore, the number of IBD-related surgical operations performed was listed.

### 2.2. Measured Radiation Exposure

The number and type of each imaging study (X-ray, CT, and MRI) conducted were listed for each patient. CED was estimated for each patient using standardized tables, as shown in Table 1 [5,9,10]. The CED for each patient was calculated by multiplying the effective dose for each procedure by the total number of procedures. To quantify an accurate total diagnostic radiation exposure, all imaging studies performed for non-IBD indications were also included. A high dose of radiation was defined as CED ≥ 50 mSv.

### 2.3. Statistical Analysis

We compared the mean doses of CED in CD and UC patients. We assessed the link between the character of diseases and absorbed radiation. CED values were correlated as well with the individual parameters in particular groups. Statistical analyses were performed using Microsoft Excel software. Individual values are presented as a number, a percentage, or mean values with standard deviation and compared using the t-test (normally distributed data) or the Mann–Whitney U-test (non-normal distribution). *p* ≤ 0.05 were considered statistically significant. To compare mean prevalence differences between groups, the χ² test (categorical variables) was performed. Kruskal–Wallis test was used to evaluate the differences between three or more independent samples. Correlations between normally distributed data were calculated using the Pearson correlation coefficient. For nonparametric data series, the Spearman correlation coefficient was used.

## 3. Results

The retrospective analysis was conducted. IBD patients were divided into two groups (CD and UC) that were statistically homogenous. The characteristics of studied patients is presented in Table 2. The mean duration of disease did not differ significantly (CD 10.72 years vs. UC 12.84 years; *p* = 0.16). Patients with CD were statistically more frequently admitted to the hospital than patients with UC. However, there was no significant difference in the mean length of a single hospitalization. A total of 40 patients with CD underwent a total of 93 CD-related surgeries, while 10 patients with UC underwent proctocolectomy.

Ileal disease location was described in 7 CD patients, colonic in 21, ileocolonic in 32. Based on the hospital charts, the location of the disease was unable to be described in three CD patients. A total of 27 CD patients presented non-stricturing/non-penetrating disease, 21 stricturing, and 17 penetrating. Seven UC patients had ulcerative proctitis, 32 had left-sided colitis, and 55 had extensive colitis (changes proximal to splenic flexure). Based on the hospital charts, the location of the disease was unable to be described in four UC patients.

The list of indicated pharmacotherapy was made. A total of 57 patients with CD received aminosalicylates, 23 immunomodulating drugs, 15 corticosteroids, and 10 anti-TNF agents. A total of 90 patients with UC received aminosalicylates, 34 immunomodulating drugs, 29 corticosteroids, and 18 anti-TNF agents.

Mean CED in the CD patients was significantly higher compared to UC patients (19.20 mSv vs. 6.66 mSv; *p* < 0.00001). CED values from particular imaging studies for individual patients are presented in Figure 2 and Figure 3. Only 1.84% of the patients received a potentially harmful dose ≥ 50 mSv, one with CD and two with UC.

Mean CED was also assessed depending on CD locations and behavior and UC extension, which is presented in Figure 4. We noticed that the localization of CD has no influence on CED. However, patients with structuring and penetrating disease absorbed significantly more radiation compared to patients with non-stricturing/non-penetrating disease. Exposure was significantly lower among patients with ulcerative proctitis and left-sided colitis than among those with extensive colitis.

In this study, significantly more imaging studies were conducted in the group of patients with CD compared to the patients with UC (250 vs. 233 *p* = 0.0002, respectively). A total of 73% of all imaging studies performed exposed patients to a dose of ionizing radiation, statistically more in CD patients, compared to UC patients (*p* = 0.007). The predominant disproportion of conducted examinations in this particular group of patients concerned CT enterography. This study was performed almost three times more often in CD patients than in the UC group (63 vs. 22; *p* < 0.00001, respectively). MRI were more often conducted in patients with CD (69 studies) compared to 63 in UC (*p* = 0.026). Imaging studies performed are presented in Table 3.

The correlations of individual parameters with CED are presented in Table 4. We have identified three factors associated with CED in the whole study group of IBD patients: number of hospitalization, length of hospitalization and number of surgical procedures. Length of hospitalization as well as number of surgical procedures correlates with CED in the whole study group as well as in subgroups of CD and UC patients. Age is a factor that correlates positively with CED in patients with CD; however, this is not observed in UC patients. We did not observe an association between the duration of the disease as well as steroid treatment and CED.

## 4. Discussion

The presence of CD is the most important factor of exposure to radiation among patients with IBD. In this study, they obtained a much higher dose of radiation than patients with UC. The range of mean CED in available studies was reported as 14.3–27.5 mSv and 3.65–6.8 mSv for CD and UC patients, respectively [3,11,12,13], which is consistent with results of our study. Higher radiation exposure among CD patients compared to patients with UC represents the biology of the disease. The extent of the disease frequently includes the small intestine, which is not easily accessed by endoscopy. Therefore, imaging examinations are helpful tools for the CD diagnosis and monitoring the disease activity next to the endoscopy. Imaging studies can evaluate fistulas, abscesses, and strictures, which are common complications in CD [14].

CD patients with strictures or penetrating disease were more likely to receive higher radiation doses. This outcome was also described in many other studies [3,15,16,17,18]. The occurrence of such complications forces the patients to visit a physician who orders imaging tests to determine the cause of the complaints. Abscesses and fistulas are often not detectable by endoscopy. The presence of the stricture frequently makes endoscopy impossible to perform, making imaging necessary. Such patients also often require surgery, which relates to more frequent imaging as well.

Localization of CD has no influence on CED, which has been confirmed in other studies [16,17]. Inflammation caused by CD can involve any segment of the gastrointestinal tract. The course of CD is not stable. Over time the new lesion may appear in another segment [2]. Therefore, patients usually have diagnostics of the entire gastrointestinal tract during follow-up visits. On the other hand, localization in UC has impact on CED. Patients with extensive colitis absorbed markedly more radiation than patients with ulcerative proctitis or left-sided colitis. A similar result was described in previous study [19].

In this study, we have demonstrated that number of performed IBD-related surgeries, and number and length of hospitalizations are correlated with CED among patients with IBD. Association between multiple surgeries and a higher dose of radiation was indicated in previous studies [3,16,17]. The necessity of assessment before and after surgical procedures in order to monitor post-operative progress or complications leads to multiple imaging studies. Number and length of hospitalizations and necessity of surgeries can be a marker of active disease. Exacerbations are the cause of more frequent and longer hospitalizations, during which diagnostic imaging is being performed to estimate disease activity and guide further management.

We identified two characteristic features that distinguish patients with CD from patients with UC, including the necessity of intestinal surgery for underlying disease and an increased number of hospitalizations per person. Patients with CD undergo surgeries more often than patients with UC. CD is a disease, where an operation is not curative, therefore minimally invasive surgery is recommended. Patients need re-operation very often [2]. A number of interventions entail a number of hospitalizations, which may be one of the reasons for more frequent admission to the hospital. Lesions in UC appear only in the large intestine, therefore restorative proctocolectomy with ileal pouch-anal anastomosis is recommended surgery that cures a patient [1]. 

In CD patients, correlation between CED and length of hospitalization is observed as well as in the whole group of IBD patients. A subsequent factor which is positively correlated with CED is age. The consistent result was observed also by Levi et. al [20]. Authors suggest that physicians may be more indulgent in ordering imaging studies with ionization radiation in older patients because of the reduced risk for developing cancer compared to younger patients. Physicians may also consider studies without radiation in younger individuals because of their childbearing potentials. However, other studies did not demonstrate any association between age and radiation exposure [21,22].

Corticosteroid use is commonly reported as a predictor of higher radiation exposure [16,19,23]. Patients with active or complicated disease are more likely to receive corticosteroids and undergo diagnostic imaging to direct therapeutic decisions [23]. However, recent analysis performed by Patil et al. revealed that corticosteroids reduce radiation dose from diagnostic imaging after the year following initiation of therapy when compared to the preceding year. This study demonstrated also that one year of anti-TNF therapy is related to significantly lower overall imaging examinations and reduced CED [24]. In our analysis, the groups of medications taken were listed. A total of 23% of patients with CD were taking corticosteroids. In our study, we did not notice a correlation between the type of treatment and CED.

Only 1.84% of patients with IBD received a potentially harmful dose ≥ 50 mSv, one with CD, and two with UC. Other studies have reported a higher frequency of exposure to high levels of ionizing radiation among patients with IBD. A recent meta-analysis estimated that about 1 in 10 IBD patients is at risk of exposure to high CED [21]. The reason why in our analysis patients did not exceed the CED threshold can be explained by the fact that we did not perform an analysis of imaging studies from other departments. For example, Kroeker et al. [11] estimated that 35% of CT scans in IBD patients were conducted in the emergency department. Furthermore, some studies reported that patients receive a higher dose of radiation at the onset of their disease. Nguyen et al. [14] demonstrated that CD patients were more than 2-fold more likely to receive > 100 mSv of cumulative ionizing radiation during the first 5 years following the diagnosis. Englund et al. [25] showed that individuals with CD were exposed to greater doses of ionizing radiation one year before diagnosis than three years after diagnosis. Mean CED declines with the duration of the disease. The mean duration of the disease in our patients was about 10 years, which can be a cause of the low percentage of patients exposed to CED dose ≥ 50 mSv. The period of early disease with possible higher exposure to radiation was probably not included in the study. That can be also a reason why we did not notice a correlation between the length of disease and CED.

Imaging tests with the patient’s exposure to ionizing radiation were selected more often. Over 75% of absorbed ionizing radiation in CD patients came from CT enterography. CT scans have advantages: rapid imaging, availability, and lower cost. However, their major disadvantage is ionizing radiation and therefore increased risk of malignancy [26]. Moreover, there are no validated scales on intestinal activity based on CT enterography, whereas several indexes have been developed for evaluating the CD activity in MRI i.e., Magnetic Resonance Index of Activity, Acute Inflammation Score, Clermont Score, and Sailer Index. According to ECCO-ESGAR recommendation, MRI is the study of choice in IBD patients [27]. MRI and US should be more frequently considered especially in young individuals in the reproductive period. All of these procedures have similar high sensitivity and specificity for the diagnosis of IBD [28]. Fiorino et al. [29] conducted a prospective study comparing the sensitivity, specificity, and accuracy of CT enterography and MRI in patients with ileocolonic CD. They demonstrated that MRI and CT enterography are equally valid in the estimation of disease activity and associated complications. MRI is statistically more advantageous in evaluating intestinal strictures and ileal wall enhancement compared to CT. Therefore, the method with lower potential risk for the patient should be chosen [29]. The advantages of MRI include lack of radiation and improved soft tissue resolution, especially for perianal disease and fistulas. Nevertheless, MRI has limitations such as a long time of imaging, higher cost, and limited accessibility. Moreover, gadolinium may be retained in the brain of patients who undergo MRI [27]. An alternative or complementary technique for the MRI is the small bowel US. It allows for dynamic real-time bowel assessment. This procedure is commonly available and radiation-free. It can be used to differentiate between CD and UC and to detect related complications. Nevertheless, it is an operator- and patient-dependent method [30,31].

During these 5 years of research, no one developed malignancy. However, the time of the follow-up was short. Therefore, the three patients who received CED > 50 mSv remain under our special concern.

This study has limitations. It was conducted on a small group of patients. We considered only the list of imaging studies performed in the Gastroenterology and Hepatology Department. Moreover, we did not analyze indications for procedures performed. As a consequence, we cannot correlate these parameters with CED. However, we consider our study as preliminary work, and plan to continue research on a higher number of patients, and with longer follow-up. We believe it will allow us to disclose possible complications of ionization radiation.

In conclusion, our work shows exposure to ionizing radiation among IBD patients over 5 years and factors associated with the occurrence of higher radiation doses in such patients. The presence of CD is connected to higher exposure to radiation among patients with IBD. The number of surgeries, and number and total length of hospitalizations correlate with CED among IBD patients. CD patients with strictures or penetrating disease and UC patients with extensive colitis were more likely to receive higher radiation doses.

We suggest that, concerning harmful effects of radiation, non-ionizing techniques (MR and US) should be considered more often.

## Figures and Tables

**Figure 1 diagnostics-11-02387-f001:**
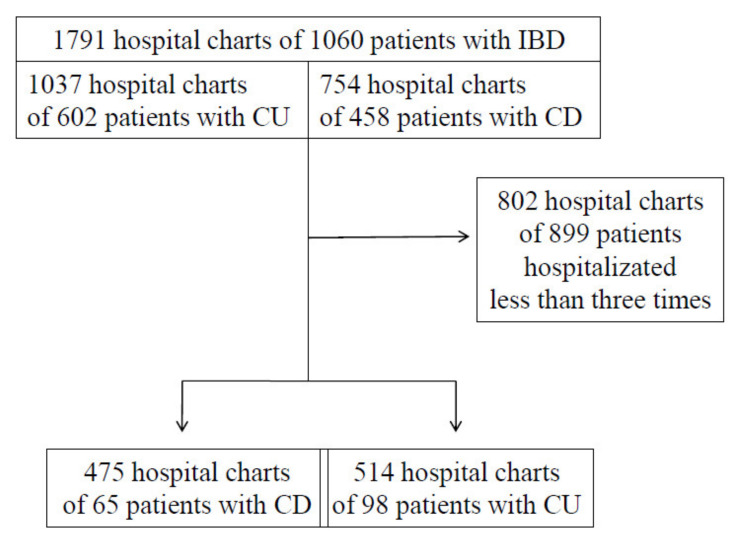
Flowchart for patient inclusion.

**Figure 2 diagnostics-11-02387-f002:**
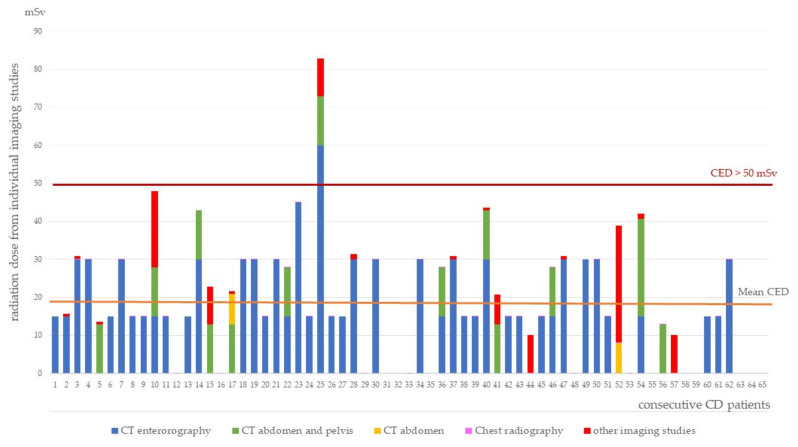
Cumulative effective dose in patients with Crohn’s disease.

**Figure 3 diagnostics-11-02387-f003:**
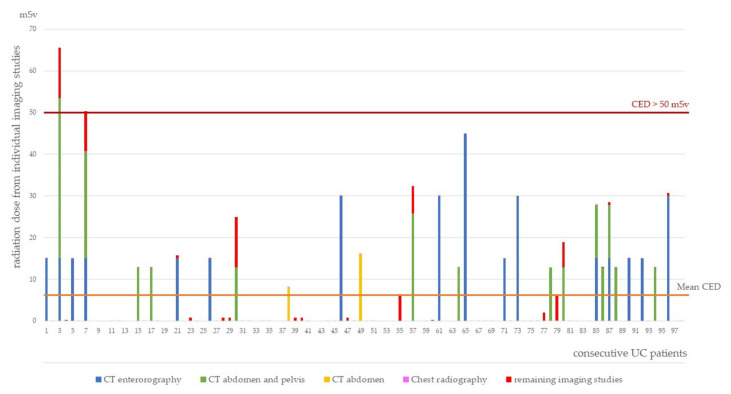
Cumulative effective dose in patients with ulcerative colitis.

**Figure 4 diagnostics-11-02387-f004:**
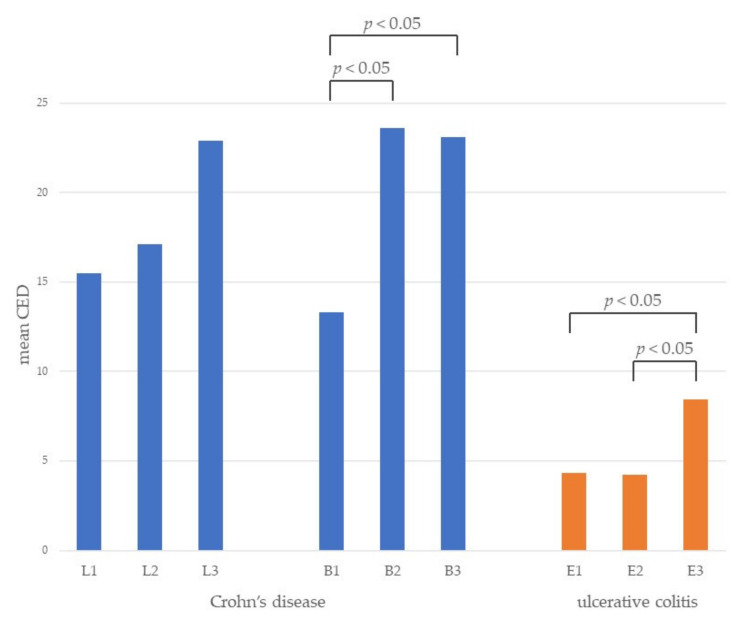
Mean cumulative effective dose (CED) in different location and behavior of Crohn’s disease and extension of ulcerative colitis. L1—ileal, L2—colonic, L3—ileocolonic, B1—non-stricturing/non-penetrating, B2—stricturing, B3—penetrating, E1—ulcerative proctitis, E2—left-sided colitis, E3—extensive colitis, *p* < 0.05 (B1 to B2 and B3; E3 to E1 and E2).

**Table 1 diagnostics-11-02387-t001:** Effective doses for most commonly performed diagnostic radiology procedures.

Type of Procedure	Effective Dose, mSv	Reference
CT enterorography	15	[9]
CT abdomen and pelvis	12.8	[10]
CT abdomen	8	[5]
Abdomen radiography	0.7	[5]
Chest radiography	0.02	[5]
Endoscopic retrograde cholangiopancreatography	4	[5]

Abbreviations: mSv, millisieverts; CT, computer tomography.

**Table 2 diagnostics-11-02387-t002:** Characteristics of the studied patients.

Features	CD	UC	*p*-Value
Females/males, n	33/32	47/51	0.85
Mean age (SD)	40.8 (14.55)	42.65 (15.66)	0.45
Mean duration of the disease, years (SD)	10.72 (8.74)	12.84 (9.769)	0.16
Mean age of the onset of the disease, years (SD)	30.18 (14.49)	29.91 (13.18)	0.91
Number of hospitalizations of all patients, n	475	514	0.02
Hospitalization days of all patients	1879	2572	0.37
Mean length of single hospitalization, days (SD)	5.63 (4.069)	5.35 (3.00)	0.31
Number of patients after surgical procedures in course of disease, n (%)	40 (61.54%)	10 (10.20%)	<0.001
Location of CD L1/L2/L3 (n)	7/21/34		
Behavior of CD B1/B2/B3 (n)	27/21/17		
Extension of UC E1/E2/E3 (n)		7/32/55	

Abbreviations: CD, Crohn’s disease; UC, ulcerative colitis; L1—ileal, L2—colonic, L3—ileocolonic; B1—non-stricturing/non-penetrating, B2—stricturing, B3—penetrating, E1—ulcerative proctitis, E2—left-sided colitis, E3—extensive colitis.

**Table 3 diagnostics-11-02387-t003:** Number of imaging studies performed.

Number of Imaging Studies	CD	UC	*p*-Value
Number of all imaging studies performed, n	250	233	<0.001
Number of imaging studies with ionization radiation performed, n	181	170	0.002
CT abdomen, n	2	3	
CT abdomen and pelvis, n	14	18	0.99
CT enterography, n	63	22	0.68
Chest X-rays, n	75	92	<0.001
Abdomen X-rays, n	11	11	0.14

Abbreviations: CD, Crohn’s disease; UC, ulcerative colitis; CT, computer tomography; MRI, magnetic resonance imaging.

**Table 4 diagnostics-11-02387-t004:** Correlation of individual parameters with cumulative effective dose.

Parameters	All Patients		CD		UC	
	r	*p*-Value	r	*p*-Value	r	*p*-Value
Age	0.15	0,07	0.34	<0.01	0.07	0.51
Duration of the disease	0.14	0.07	0.01	0.92	0.07	0.48
Number of hospitalizations	0.19	0.02	0.11	0.40	0.18	0.09
Hospitalization days	0.49	<0.00001	0.55	<0.00001	0.48	<0.00001
Number of surgical procedures	0.40	<0.00001	0.27	0.03	0.29	<0.01
Steroid treatment	−0.05	0.56	0.04	0.77	−0.11	0.28

Abbreviations: CD, Crohn’s disease; UC, ulcerative colitis.

## Data Availability

Not applicable.

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
