# Peer review of "Cumulative Effective Dose from Medical Imaging in Inflammatory Bowel Disease"

_diagnostics, 2021, doi:10.3390/diagnostics11122387_

Round 1

Reviewer 1 Report

The manuscript shows a retrospective study of cumulative effective dose from medical imaging in inflammatory bowel disease. The results show that few patients received high levels of CED and that imaging studies for these pathologies are mainly based on the usage of ionizing radiation.

Though the results may resonate for medical doctors and radiographers professionals, there are some weaknesses that need to be addressed:

  1. UC and CD have different pathophysiologic processes which translated in different imaging patterns. Moreover, also the treatment strategies differ. This means that the results in terms of type of imaging studies and CED values are per disease different. This needed segmentation and relation in between variables are lacking from this study and in my opinion may compromise the conclusions of it. For example, it is standard that the patient performs a chest x-ray prior to a surgical procedure. Was this taken into account? CT enterography is more common for CD patients, was this finding correlated with the clinical question and clinical patterns? Does it justify the Effective Dose values - Which in radiology stands for the justification principle.
  2. Figure 2 and 3 should have on x axis the time between exams or the disease timeline. With this the reader could have a more precise evaluation on how CED growths for these patients. Also the dots in the graph could be color coded based on the type of imaging study. It is hard to follow a cumulative process when there is now data regarding time and number and type of exams per patient throughout the studied time. 
  3. Disease severity, patient biotype and CT protocols should also be included in order to better sustain the results and the presented discussion. Otherwise statements like "Only 1.84% of patients with IBD received a potentially harmful dose (...)".  

To sum up, the manuscript needs to include clinical and exam data so that the discussion could be more sound. 

Author Response

We would like to thank Reviewer for the broad and valuable review.

Here are the answers for the remarks:  

1. UC and CD have different pathophysiologic processes which translated in different imaging patterns. Moreover, also the treatment strategies differ. This means that the results in terms of type of imaging studies and CED values are per disease different. This needed segmentation and relation in between variables are lacking from this study and in my opinion may compromise the conclusions of it. For example, it is standard that the patient performs a chest x-ray prior to a surgical procedure. Was this taken into account? CT enterography is more common for CD patients, was this finding correlated with the clinical question and clinical patterns? Does it justify the Effective Dose values - Which in radiology stands for the justification principle.

Thank the reviewer for these value remarks. We have looked into the hospital charts of gastroenterology department. If procedures were mentioned in the charts or in the electronic hospital  system they were added. Procedures performed in other hospitals might had be missed for analysis. We have included clinical date of the patients: localization of the disease and the behavior of the Crohn’s disease and the cumulative effective dose was compared in these subgroups. To quantify an accurate total diagnostic radiation exposure, all imaging studies performed for non-IBD indications were also included.

 2. Figure 2 and 3 should have on x axis the time between exams or the disease timeline. With this the reader could have a more precise evaluation on how CED growths for these patients. Also the dots in the graph could be color coded based on the type of imaging study. It is hard to follow a cumulative process when there is now data regarding time and number and type of exams per patient throughout the studied time. 

Thank the reviewer for these remarks. We had developed Fig.2 and Fig.3. Dots in the graph was changed on bars. Color bars were implemented to visualize what and how many  medical procedures with radiation dose were performed. The x axis label was also added due to remarks of other reviewer. X axis shows consecutive patients. In our opinion, analyzing the course of the absorbed dose change in such a short time will not bring relevant data. However, we consider our study as preliminary work. We plan to continue our research on a higher number of patients, and with longer follow-up. The correlation of the CED over a longer time may show an important pattern. For example changes in imagining regarding the duration of the disease. We believe also that longer time of follow-up may disclosure some complications of ionization radiation.

3. Disease severity, patient biotype and CT protocols should also be included in order to better sustain the results and the presented discussion. Otherwise statements like "Only 1.84% of patients with IBD received a potentially harmful dose (...)".  

We have added the data about extension of the ulcerative colitis, location and behavior of Crohn’s disease (Montreal classification). The mean cumulative dose in the subgroups was compared. As one of the inclusion criteria for our study was multiple hospitalization the disease severity was altering during the time- therefore we have decided to include the disease location and behavior for comparison with cumulative effective dose. CT protocols were unable to obtain - imaging studies are performed in a different department, with complex access to their charts. However, the majority of other similar studies also used standardized tablets to determine cumulative effective doses.

4. To sum up, the manuscript needs to include clinical and exam data so that the discussion could be more sound. 

The clinical data of disease extension and behavior of Crohn’s disease was added.

Reviewer 2 Report

The authors present a paper entitled "Cumulative effective dose from medical imaging in inflammatory bowel disease".
The manuscript is well organized and well written. The work is based on a retrospective chart view of patients with inflammatory bowel diseases hospitalized between 2015 and 2019. The discussion is broad and focused on harmful effects of radiation in order to identify factors associated with exposure to high doses of diagnostic radiation. Although the work has limitations, already underlined by the authors themselves, I believe that the present manuscript can be considered a preliminary work with the possibility of extending it in the future to a higher number of patients and of making correlations with other factors or symptoms presented by patients. The most relevant aspect is the examination of the potential danger of radiation used in diagnostic imaging.

Author Response

The authors present a paper entitled "Cumulative effective dose from medical imaging in inflammatory bowel disease". The manuscript is well organized and well written. The work is based on a retrospective chart view of patients with inflammatory bowel diseases hospitalized between 2015 and 2019. The discussion is broad and focused on harmful effects of radiation in order to identify factors associated with exposure to high doses of diagnostic radiation. Although the work has limitations, already underlined by the authors themselves, I believe that the present manuscript can be considered a preliminary work with the possibility of extending it in the future to a higher number of patients and of making correlations with other factors or symptoms presented by patients. The most relevant aspect is the examination of the potential danger of radiation used in diagnostic imaging.

We would like to thank Reviewer for the kind words on our paper. Of course, this is the preliminary work, that should be extended. We plan to continue our research on a higher number of patients, and with longer follow up. We believe it will allow us to disclosure some complications of ionization radiation. At the moment, concerning the remarks of other reviewers we have included some clinical data of the patients: localization of the disease and the behavior of the Crohn’s disease.

Reviewer 3 Report

This is an important study about the CED for IBD. I would suggest to a major revision to see if the authors could address the concerns I have。

  1. the figure 2 and 3 are really confusing, since there is no axis label etc.
  2.  i would suggest to plot the CED as a function of age, number of treatment, gender etc. to make it easier for the readers to understand the data.
  3. is there any further information about the 3 patients recieving more than 50mSv?  Do they have tumor or other possible side effects?

Author Response

We would like to thank Reviewer for the remarks. Here are the answers:

1. the figure 2 and 3 are really confusing, since there is no axis label etc.

The axis label was added. Additionally color bars were implemented to visualize what exact medical procedures with radiation dose were performed.

2. I would suggest to plot the CED as a function of age, number of treatment, gender etc. to make it easier for the readers to understand the data.

Thank you for this remark. More figures were added to make the manuscript more comprehensive for the readers.

3. is there any further information about the 3 patients recieving more than 50mSv?  Do they have tumor or other possible side effects?

 During these 5 years of research, no one developed malignancy. However, the time of the follow-up is short. Therefore these three patients, who received CED >50 mSv remain under our special concern. At the moment, they are free from cancer.

Round 2

Reviewer 1 Report

Thank you to the authors for the great improvement in the manuscript. All of the questions were answered.

Reviewer 3 Report

Have no further comments.